# A Tailored MOOC Fostering Intercultural Conflict Management in the Educational Context: Evidence from Italy

Elena Dell'Aquila [1,2,*], Federica Vallone [1,3], Maria Clelia Zurlo [1,3] and Davide Marocco [1,2]

1  Department of Humanities, University of Naples Federico II, 80133 Naples, Italy; federica.vallone@unina.it (F.V.); zurlo@unina.it (M.C.Z.); davide.marocco@unina.it (D.M.)
2  Natural Artificial Cognition Laboratory "Orazio Miglino", University of Naples Federico II, 80133 Naples, Italy
3  Dynamic Psychology Laboratory, University of Naples Federico II, 80133 Naples, Italy
*  Correspondence: elena.dellaquila@unina.it

**Abstract:** Managing relationships between/with students is one of the main duties that teachers are asked to fulfill, which becomes even more challenging in multi-ethnic/multicultural educational contexts. Responding to the need for culturally-qualified training for school professionals and given the increasing use of Massive Open Online Courses (MOOCs)—often without evidence supporting their efficacy—this study will evaluate the potential of a tailored MOOC—designed to tackle overt/covert discrimination and foster inclusion and culturally responsive teacher–student interactions—to promote teachers' awareness and competence in conflict management. Overall, 206 Italian teachers experienced the MOOC, of whom 99 completed the Rahim Conflict Management Inventory-II, assessing Conflict Management Styles (Integrating, Obliging, Compromising, Dominating, and Avoiding) pre- and post-MOOC. Potential changes in the adoption of Conflict Management Styles after completing the MOOC were evaluated. After completing the MOOC, there was a reduction in teachers' recourse to the Avoiding and Obliging styles and, of note, more informed/targeted use of the Dominating style. These findings provided evidence-based contributions sustaining that the MOOC may effectively foster a more aware, engaged, and active pattern for managing relationships and conflicts within classes, thus potentially having a tangible positive impact on the real everyday life of teachers experiencing this training and their students.

**Keywords:** conflict management; ethnic relations; Massive Open Online Courses; online; teaching; teacher–student relationships; web-based tools

## 1. Introduction

Within the current superdiverse world, characterized by increasing multiculturality, new needs and challenges have emerged to effectively face cultural/ethnic differences at the individual and societal levels (Crul et al. 2023; Meissner et al. 2023). The education context is one of the chief places to implement actions which foster constructive intercultural relationships and intercept the emergence of issues and tensions due to diversity (EC 2020; Gonçalves et al. 2016).

Over recent years, the number of school-age people with migrant backgrounds in many European countries has been growing (OECD 2018a). This condition has highlighted a need to reflect on how to effectively support students with migrant backgrounds, as they are at a higher risk of underperformance in education, dropping out of school earlier, and later challenges in the world of labor than their native counterparts (OECD 2023). Due to these concerns, schools are becoming increasingly aware of the need to identify successful educational interventions which favor a supportive and nurturing learning environment to increase engagement and equity in the education process (EC 2021).

Teachers are key actors in enacting inclusive education policy, bringing actual changes in the education of students. Specifically, nowadays, teachers have new key roles and duties

within the school and society (Donlevy and Rajania 2016). Teachers need to effectively manage classes which feature cultural and ethnic diversity and help students adjust to the superdiverse school and social contexts. Teachers also need to successfully manage relationships with and between students (Zurlo et al. 2020) and secure a safe/equal/inclusive context for all of the students and their families (Hannon and O'Donnell 2022).

For this very reason, teachers, globally, are pressured by growing expectations, as they need to possess or acquire new individual and relational skills to successfully fulfill the teaching role (Ası and Joyce 2023; EC 2017; Vallone et al. 2022). Specifically, they need to possess/enhance not only their knowledge and awareness about multicultural education but also their individual skills to deal with potential conflicts which could arise in classes due to differences/mismatches in values, cultural backgrounds, and worldviews (Evans et al. 2009; Han and Han 2019; Kunemund et al. 2020). The encountering of different cultures and perspectives within the school context is a reality to be adequately addressed (Euwema and Van Emmerik 2007). This condition may result in conflicts deriving from misunderstandings, lack of knowledge about different cultural backgrounds, and a Color-blind approach to diversity (for example, when teachers/educators act as though "we are all the same, regardless of differences"). Conflicts may, however, also derive from overt and covert expressions of prejudices, discriminations, and exclusions (Anderstaf et al. 2021; Dong et al. 2008; Expósito et al. 2014; Kyere et al. 2023; Jurtikova 2013; OECD 2018b; Stevens and Dworkin 2019). The consequences of teachers failing to manage intercultural and interethnic conflicts within the classrooms are countless, and they may vary from conflict escalation within the class to the severe psychological suffering of actors involved and the development of a hostile/unsafe class climate. Thus, increasing the intercultural competence of teachers allows them to approach culturally/ethnically diverse populations appropriately.

Within this portrait, based on one of the most internationally adopted models for conflict management, namely the Rahim's Model of Conflict Management (Rahim 2001; Rahim and Bonoma 1979), research has investigated the variety of management strategies that teachers may use to deal with the different situations which potentially occur during classes both with and between students (Dell'Aquila et al. 2022; Dogan 2016; Morris-Rothschild and Brassard 2006; Vallone et al. 2022; Zurlo et al. 2020). Specifically, teachers may expend efforts and a great amount of time to carefully deepen students' perspectives and the reason for the conflict, trying to solve it effectively (i.e., Integrating style). Moreover, teachers may choose to save time—needed for achieving the learning outcomes—and quickly find a compromise to solve the issue (i.e., Compromising style). However, teachers may also opt to reach a conflict resolution which is biased towards the student's interest to safeguard the relationship with him/her (i.e., Obliging style), or to avoid dealing with the conflict/postpone any confrontation (i.e., Avoiding style). Finally, teachers may decide to use more direct means to handle conflicts, which may vary along a continuum ranging from a more control-oriented/hostile style to a firm expression of authority (i.e., Dominating style). Teachers can use this latter form, the Dominating style, to accomplish their role as class managers.

Despite the abundance of research and practical efforts that policymakers/educational key actors undertake to strive to support school professionals in the enhancement of conflict management and interethnic/intercultural competences (Malone and Ishmail 2020; Papageorgiou et al. 2023), overall, school staff still express a need for culturally-qualified training and educational tools (EC 2017; Burnell and Schnackenberg 2015; Keengwe 2010; OECD 2014). Therefore, there is a clear necessity to propose further customized contributions to effectively support teachers' professional development.

In this direction, recently, research has increasingly highlighted that digital tools and e-environments may provide several advantages to learners, i.e., overcoming the boundary of space and time; higher individualization of the learning experience; innovative, interactive, and engaging environments; cost-effectiveness advantages; flexibility; transferability (Almahasees et al. 2021; Fidalgo et al. 2020; Prosen et al. 2022). In the past decade, the

online learning experience has been further enhanced through the development of Massive Open Online Courses (MOOCs), which have indeed made education even more easily accessible to people globally (Hew and Cheung 2014; Hone and Said 2016; Jo 2018).

Accordingly, within the project ACCORD, and in line with the national guidelines for MOOC preparation (Susanna 2017; Farrow 2019), a MOOC was developed and embedded within a tailored e-platform. The MOOC was designed to train educators, teachers, and school professionals to handle diversity and deal with conflicts that can occur in classrooms through a freely available online multicultural training course. In line with the EU Digital Education Action Plan (2021–2027) and the AGENDA 2030, the MOOC responds to the need for innovative learning approaches to reduce disparities in accessing and engaging with formal and non-formal education.

### 1.1. MOOC Contents

The MOOC proposed in the present study has been structured considering both the pros and cons reported in the literature for the design and implementation of MOOCs (Gamage and Kalansooriya 2021). According to our experience, and in line with findings in this research field (Lee et al. 2020), the most influential dimensions for a successful MOOC are represented by: self-regulated learning strategies; a custom-made system of feedback (Dell'Aquila et al. 2017); the adoption of intelligent technology that enables creating tailored learning pathways for different users (Papadimitriou 2023).

The MOOC consists of different contents, namely, Lessons, Serious Games, and Self-assessment questionnaires (Figure 1). The five lessons provide teachers with in-depth multimedia learning material that is freely available online (further information links and videos, slides, and scripts). The lessons deal with the topics specified in Appendix A Table A1. The MOOC also includes a Serious Game to enrich and increase users' engagement in the personal experience of interethnic conflicts. The Serious Game is a single-player tool to train intercultural communication and conflict management skills through realistic scenarios during interaction with artificial agents. The game simulates a dialogue between two 3D avatars (i.e., one represents the teacher and is controlled by the user, and the other represents the student and is controlled by the computer). The avatars perform basic expressions based on verbal and non-verbal components (e.g., facial expressions and body gesture) that model the main dimensions of the identified relevant psychological and pedagogical theories of Effective Communication (Dryden and Constantinou 2004) and Conflict Management Styles (Rahim and Bonoma 1979). The game allows the user to play five different scenarios developed to convey a particular type of interethnic conflict in the classroom context (see Appendix A Table A2). At the end of the gaming experience, teachers can benefit from a customized debriefing about the negotiation path played during the conflict scenarios. The assessment system allows teachers to become aware of personal conflict handling styles and related skills which are particularly relevant to successfully managing multicultural classes.

Finally, the MOOC also comprises "Self-assessment Tools". The assessment path of the e-platform includes questionnaires to: (1) investigate the users' conflict management skills; (2) assess personal experiences regarding the learning experience undertaken through the game scenarios which are related to the Conflict Management Styles and multicultural dimensions involved. The tools allow for assessing the users' training achievements and perceived learning development after the completion of the MOOC.

The e-platform is sustained by rigorous technological (i.e., EduTechRPG; Simulation Technology-drama based system; Dell'Aquila et al. 2017, 2020; Marocco et al. 2019), psychological (Vallone et al. 2019), and pedagogical frameworks (Frossard and Barajas 2018). Nonetheless, there is a need to empirically prove the validity of the MOOC, as mentioned above. Indeed, despite the rapid growth in the interest in e-learning approaches over recent years, more studies are needed to evaluate the validity of online tools in educational settings (Hewson and Charlton 2019; Soffer et al. 2017).

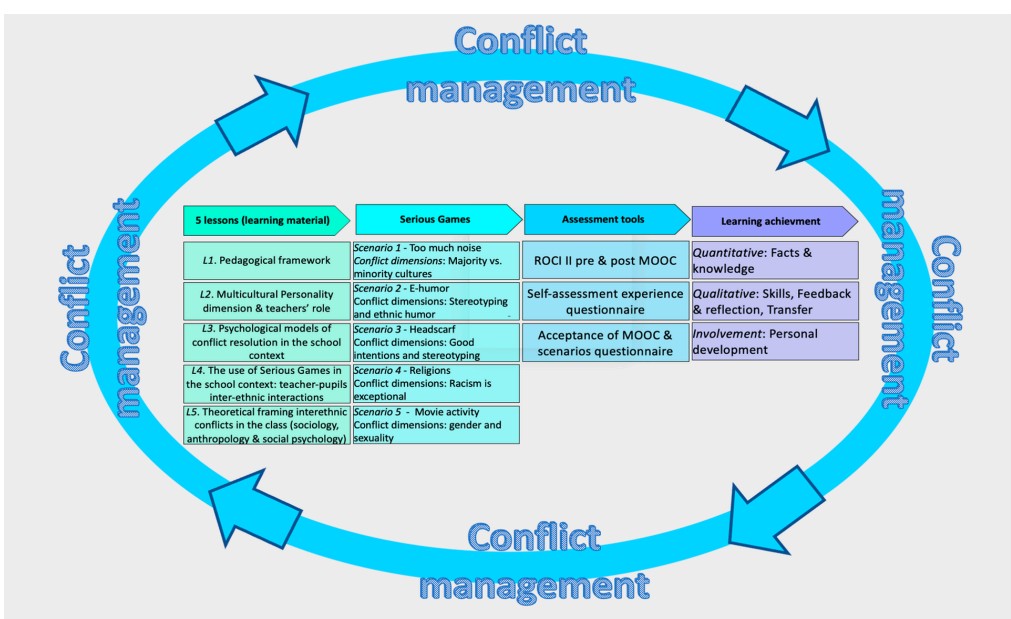

**Figure 1.** The ACCORD MOOC structure.

## 1.2. The Present Study

The present study aims to evaluate the potential of a tailored MOOC—specifically designed to tackle overt/covert discrimination and foster inclusion and culturally responsive teacher–student interactions—to effectively promote teachers' awareness and adaptiveness in conflict management. In particular, the study aims to evaluate the effectiveness of the MOOC in producing significant changes in the Conflict Management Styles adopted by teachers.

In our expectations, we were confident that the MOOC could foster not only knowledge and learning achievements but also actual modifications in latent dimensions, such as interpersonal styles. Several teachers may indeed perceive that they do not have the experience or knowledge to effectively manage culturally and ethnically diverse classes. This feeling may also be partially due to teachers' lack of awareness of their individual and relational characteristics, skills, and resources.

In other words, by offering a valid online tool, the current study sought to provide a practical answer to a key issue regarding the current educational context, namely how to effectively support educational actors in the rapid shift from a traditional mindset—that favors "optimal learning outcomes" within homogeneous educational contexts—to a multifaceted approach—that attempts to simultaneously promote both valuable learning outcomes and satisfactory relationships within the current superdiverse educational context.

## 2. Materials and Methods

### 2.1. Participants and Procedure

The MOOC was made available online as part of the ACCORD Project. The implementation of the MOOC was preceded by a preliminary piloting for testing the usability of the e-platforms (lessons, learning material, and questionnaires). This phase was extremely important before the MOOC was released, considering that the founding methodology of the MOOC is designed to allow users to train themselves on the topic autonomously and from home. In the case of any impediment, teachers could not receive immediate feedback and/or support (available via an inquiry/help automatic button). The pilot study was conducted with 50 secondary school teachers of the ACCORD Project country consortium by administering both paper-based prototypes and the online version of the MOOC. The pilot phase was crucial to identifying any potential issue that could impact the effectiveness of the learning experience (e.g., content-related issues, technical issues, usability problems,

issues in the basic game and scenarios design, and problems with the artificially intelligent tutor). We used the collected feedback to refine and improve the final version of the MOOC. Afterwards, we spread the final version of the MOOC online through formal channels (e.g., institutional mailing lists) and informal channels (e.g., word-of-mouth and social media). There were no inclusion/exclusion criteria for enrolling in the MOOC, except for the condition of being a school teacher.

Overall, 206 Italian teachers enrolled and completed the online training courses on a voluntary basis; of those, 171 completed the questionnaire at the beginning of the learning experience (pre-MOOC). However, 99 teachers (Age Mean = 47.79, SD = 8.35, Range = 24–65; Working Seniority Mean = 18.19, SD = 9.68, Range = 0–43) fully completed the questionnaire both pre- and post-MOOC and, therefore, were included in the final dataset. Thus, there were no missing data. The characteristics of the participants are reported in Table 1. All of the procedures were performed in accordance with the 1964 Helsinki declaration and its later amendments or comparable ethical standards. The study was approved by the Ethical Committee of Psychological Research (CERP) of University of Naples Federico II (Protocol Code: 36/2019; Approval Date: 28 November 2019).

**Table 1.** Teachers' background information.

| | Value<br>[*n* (%)] |
|---|---|
| **Gender** | |
| Women | 85 (85.9) |
| Men | 14 (14.1) |
| **Age** | |
| <35 years | 7 (7.1) |
| 35–45 years | 29 (29.3) |
| >45 years | 63 (63.6) |
| **Working Seniority** | |
| <5 years | 14 (14.1) |
| 5–10 years | 10 (10.1) |
| >10 years | 75 (75.8) |

*2.2. Measures*

Teachers were asked to complete a background information form (Age, Gender, Working Seniority) along with the Rahim Organizational Conflict Inventory-II (ROCI-II; Rahim 2001; Italian version: Majer 1995) before and after completing the MOOC. In the present study, we used Form B of the tool, measuring the management styles used by individuals to deal with conflicts with subordinates (i.e., students in our case). Teachers were asked to imagine/recall an intercultural conflict that occurred within classes with and/or between students, and to select the ways in which they have handled/would have handled it.

Specifically, the Rahim Organizational Conflict Inventory-II consists of 28 items (5-point Likert scale ranging from one = Strongly Disagree to five = Strongly Agree) divided into five subscales, namely: Integrating (7 items, e.g., "I try to integrate my ideas with those of my students to come up with a decision jointly"); Compromising (4 items, i.e., I usually propose a middle ground for breaking deadlock"); Dominating (5 items, e.g., "I am generally firm in pursuing my side of the issue"); Obliging (6 items, e.g., "I usually accommodate the wishes of my students"); Avoiding (6 items, e.g., "I try to stay away from disagreement with my students").

For interpreting the data, higher scores represent higher recourse to those strategies. Rahim's description of the five styles derives from the different combinations of two main dimensions, namely "Concern for Self" (i.e., the degree to which people aim to satisfy their own concerns throughout conflict management processes) and "Concern for Others" (i.e., the degree to which people aim to satisfy the concerns of the other party throughout

conflict management processes). Accordingly, the scores from the ROCI–II can be discussed according to two macro-dimensions, namely the Integrative and Distributive Dimensions.

The Integrative Dimension (ID) represents the amount of satisfaction (high–low) about the concerns received by both parties (self and others) and covers the Integrating Style and Avoiding Style. Indeed, when using the Integrating style, people attempt to increase the satisfaction of the concerns of both parties by finding solutions to the problems that are mutually acceptable, whereas, conversely, the Avoiding style leads to a reduction in satisfaction of the concerns of both parties and results in a failure to solve the issue.

In contrast, the Distributive Dimension (DD) represents the amount of satisfaction (high and low) about the concerns received by only one of the parties (self or others) and covers the Dominating Style and Obliging Style. Indeed, when using the Dominating style, people attempt to obtain a high satisfaction of concerns for self (and low for others), whereas, conversely, the Obliging style leads to low satisfaction of the concerns for self (and high satisfaction of concerns for others). In the middle ground, the Compromising style represents the point of intersection of the two dimensions, which represents an intermediate position where both parties receive a moderate level of satisfaction of their concerns from the resolution of their conflicts.

The ROCI-II has been internationally adopted, and its psychometric soundness is well-recognized (Bilsky and Wülker 2000; Munduate et al. 1999; Rahim et al. 2002; Vallone et al. 2022; Yu and Chen 2008).

*2.3. Statistical Analysis*

Data were analyzed by SPSS software (Version 27) (IMB, Chicago, IL, USA). Firstly, Descriptive statistics were conducted (Mean and Standard Deviations) for Conflict Management Styles (both pre- and post-MOOC) and a comparison was drawn with respect to the corresponding normative values reported in the Italian validation study (Majer 1995) by using Student's *t*-test analyses ($p < 0.05$). Afterwards, Repeated Measures ANOVA (Greenhouse-Geisser correction method; Wilk's Lambda [λ]; $p < 0.05$) was used to assess the effectiveness of the MOOC in terms of potential changes in the recourse to Conflict Management Styles by teachers. Finally, considering the inherent specificities of Massive Online Open Courses (MOOC), which should be designed to be broadly accessible and widely enjoyable by the target population (i.e., teachers in our case), and which—mainly—should be effective beyond individual differences, a further set of Repeated Measures ANOVA was also conducted to explore any potential changes in the recourse to Conflict Management Styles according to gender, age, and working seniority.

**3. Results**

Table 1 displays teachers' background characteristics (*n* = 99).

Table 2 shows the Means/Standard deviation scores of Conflict Management Style by teachers, pre- and post- the completion of the MOOC, together with normative data and findings from Repeated Measures ANOVA.

**Table 2.** Means (M) and Standard Deviations (SD) for Conflict Management Styles: Normative data and Repeated Measures ANOVA for pre- and post-MOOC by teachers.

| Conflict Management Styles | Normative Data | Pre MOOC | Post MOOC | Repeated Measures ANOVA | |
| --- | --- | --- | --- | --- | --- |
| | M (SD) | M (SD) | M (SD) | Wilk's Lambda | Chi-Square |
| Integrating | 4.15 (0.63) | 4.50 (0.44) | 4.46 (0.48) | 0.995 | 0.495 |
| Compromising | 3.18 (0.81) | 3.74 (0.81) | 3.65 (0.69) | 0.992 | 0.771 |
| Avoiding | 3.01 (0.86) | 3.04 (0.68) | 2.51 (0.66) | 0.733 *** | 35.75 *** |
| Obliging | 3.39 (0.66) | 2.71 (0.49) | 2.55 (0.45) | 0.953 * | 4.81 * |
| Dominating | 2.91 (0.94) | 1.80 (0.69) | 2.06 (0.79) | 0.994 * | 5.76 * |

*** $p < 0.001$; * $p < 0.05$.

With respect to the comparison with normative values and considering the Integrative Dimension (ID), teachers who accessed the MOOC adopted the Integrating style to a significantly greater extent than the normative group of workers (pre-MOOC $t = 5.43$, $p < 0.001$; post-MOOC $t = 4.80$ $p < 0.001$). Moreover, they also adopted the Avoiding style with a significantly lower extent than the normative group only after completing the MOOC (pre-MOOC $t = 0.34$, $p = 0.734$ post-MOOC $t = 5.66$, $p < 0.001$).

Considering the Distributive Dimension (DD), teachers adopted the Dominating style (pre-MOOC $t = 11.53$, $p < 0.001$; post-MOOC $t = 8.78$, $p < 0.001$) and Obliging style (pre-MOOC $t = 10.06$, $p < 0.001$; post-MOOC $t = 12.45$, $p < 0.001$) to a significantly lower extent than the normative group.

Additionally, teachers adopted the Compromising style significantly more than the normative group of workers (pre-MOOC $t = 6.64$, $p < 0.001$; post-MOOC $t = 5.63$, $p < 0.001$).

With respect to changes in the recourse to Conflict Management Styles by teachers after the completion of the MOOC, the data revealed a statistically significant reduction in the recourse to the Avoiding style ($p < 0.001$) and Obliging style ($p = 0.031$), and a statistically significant increase in the recourse to the Dominating style ($p = 0.018$). No changes in the recourse to the Integrating style ($p = 0.483$) and Compromising styles ($p = 0.382$) were found (Figure 2).

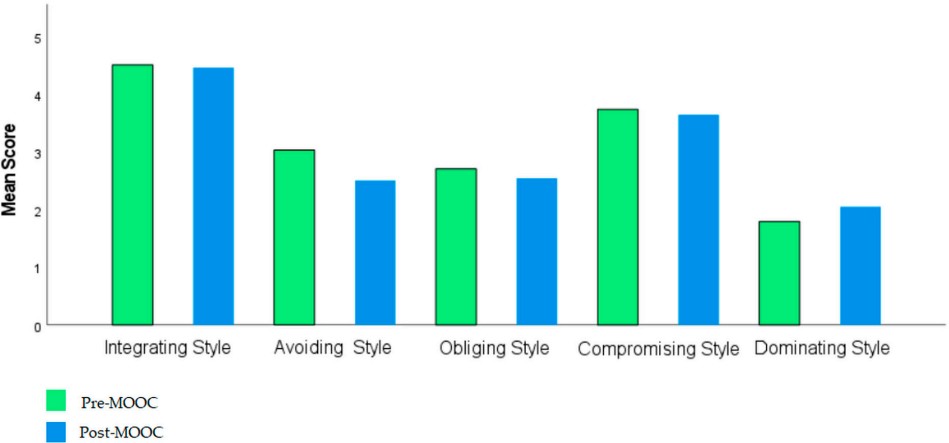

**Figure 2.** Changes pre-to-post-MOOC in conflict management strategies.

Furthermore, findings revealed no significant changes in the recourse to conflict management style by background information (i.e., *Gender* Integrating style $\lambda = 0.993$, $p = 0.417$; Compromising style $\lambda = 0.998$, $p = 0.681$; Avoiding style $\lambda = 0.985$, $p = 0.226$; Obliging style $\lambda = 0.983$, $p = 0.202$; Dominating style $\lambda = 0.999$, $p = 0.742$; *Age* Integrating style $\lambda = 0.973$, $p = 0.273$; Compromising style $\lambda = 0.998$, $p = 0.897$; Avoiding style $\lambda = 0.994$, $p = 0.736$; Obliging style $\lambda = 0.970$, $p = 0.231$; Dominating style $\lambda = 0.969$, $p = 0.223$; *Working Seniority* Integrating style $\lambda = 0.992$, $p = 0.665$; Compromising style $\lambda = 0.987$, $p = 0.540$; Avoiding style $\lambda = 0.978$, $p = 0.340$; Obliging style $\lambda = 0.969$, $p = 0.221$; Dominating style $\lambda = 1.00$, $p = 0.999$).

## 4. Discussion

This study aims to respond to the need to prove the validity of MOOCs in effectively producing changes in learners—not only in terms of increased knowledge but also in terms of deeper consciousness and actual modifications in interpersonal styles. In particular, this study provided evidence endorsing the efficacy of our tailored MOOC in effectively promoting teachers' awareness and adaptiveness in conflict management within interethnic and intercultural educational contexts.

Firstly, the findings highlighted that—beyond the MOOC experience—teachers who participated in the study reported scores reflecting a more competent use of all five Conflict Management Styles when compared with the normative group, the latter comprising a

sample of general Italian workers employed in public and private organizations (Majer 1995). These data are, however, in line with the ongoing transformations of the educational context; indeed, according to the European agenda, nowadays, teachers are increasingly demanded to promote mutual understanding, respect, and integration among students, rather than reaching high-quality learning outcomes alone (Ası and Joyce 2023; EC 2017, 2020). They need to favor a satisfactory class climate, manage relationships with and between students, and deal with the potential conflicts that could occur during classes, also due to the increasing differences in values, worldviews, and cultures in our society (e.g., Jackson and O'Grady 2019; Loode 2011; Meissner et al. 2023).

In the present study, the values gathered among teachers for the Integrating style (Integrative Dimension; ID) as well as for the Compromising style were significantly higher than the normative values even before they were involved in the MOOC experience, thus highlighting the already increased awareness of adopting these styles favorably to manage conflicts effectively (Ciuladiene and Kairiene 2017; Yu and Chen 2008). No changes between the values of pre- and post-MOOC were found, yet we did not intend to achieve this result.

In contrast, considering the Avoiding style (Integrative Dimension; ID), our findings revealed that teachers adopted this management strategy to a significantly lower extent than the normative group only after completing the MOOC. We believe that this finding is particularly valuable when considering the evidence highlighting that Avoiding is a rather common strategy adopted by teachers. This is particularly true when the topic concerns difficult cultural subjects and/or the other party involved is perceived as culturally different (Dogan 2016; Mahon 2009; Morris-Rothschild and Brassard 2006). However, conflicts within classes and, above all, intercultural conflicts may significantly escalate when teachers underestimate/avoid confronting them, potentially resulting in a vicious circle characterized by widespread perceived anger, frustration, marginalization, and inequities by students and their families (Jurtikova 2013; Malone and Ishmail 2020; Zurlo et al. 2020).

Considering the Distributive Dimension (DD) and, specifically, the Obliging style, our findings revealed that teachers were less likely to be accommodating/complaisant than the normative group, and this finding may reflect the inherent pivotal role of teachers as class managers. Indeed, despite the recourse to this strategy, it can also be adequate to defend relationships during the negotiation processes (Tehrani and Yamini 2020; Yu and Chen 2008), and the adoption of this strategy should be discouraged for people in charge of managing (Rahim 2001). Using an Obliging style leads people to sacrifice their own concerns to satisfy those of their subordinates (in this case, the students). From this perspective, teachers could also tend to adopt this style—in an unaware way—when they feel highly stressed and overwhelmed (Zurlo and Pes 2006; Zurlo et al. 2016, 2020), thus being unable to deal with the conflict more effortfully. Interestingly, teachers' recourse to this strategy was significantly reduced after completing the MOOC. We believe that this evidence also endorses the value of the MOOC in effectively supporting more adequate ways to deal with conflicts. This finding could also be interpreted by considering research highlighting that European teachers highly adopting the Obliging style were also more likely to possess low emotional stability, which represents a multicultural personality characteristic featured by a high tendency and ability to regulate their own and others' emotions when under pressure in intercultural/interethnic contexts (Vallone et al. 2020; Vallone et al. 2022).

Finally, of note, findings revealed that teachers adopted Dominating style (Distributive Dimension, DD) to a significantly lower extent than the normative group. However, the recourse of teachers to this strategy significantly increased after completing the MOOC (yet remaining still lower than the normative group). We consider this result as one of the key proofs that the proposed MOOC fostered awareness among teachers. Indeed, as underlined in the MOOC contents—according to Rahim' Model (Rahim et al. 2001, 2002)—each of the five strategies adopted to manage conflict can be adequate or, conversely, inadequate

according to multiple factors, such as the specific context, the conflictual situation, the other party involved, own role.

Teachers may inappropriately adopt the dominating style to aggressively impose his/her position along with depreciation, frequently without reasoning and ignoring students' expectations and needs (Dogan 2016; Morris-Rothschild and Brassard 2006; Zurlo et al. 2020). However, teachers can appropriately use this style in those situations when they need to take a quick solution, to actively battle discriminations/prejudices within classes, or when teachers must assert their reasons to deal with the conflict based on their knowledge, competences, and/or experience. In such cases, we recommend teachers to adopt this style cautiously and with self-awareness, rather than stigmatizing it. Dominating style may indeed serve to prevent, limit, and/or regulate those students' behaviors that could harm/damage others as well as him/her (e.g., distracting/damaging other students; reducing satisfaction/participation/wellbeing within classes; lowering motivation; using the classwork time unproductively).

In sum, overall findings were all united to sustain that the MOOC may have effectively fostered a more aware, engaged, and active pattern for managing conflicts within classes, with particular emphasis on intercultural and interethnic conflicts, thus potentially having a tangible positive impact on the real everyday relational life of teachers experiencing this training and their students.

Despite these findings, this study had some limitations that should be addressed. Firstly, self-report measures were used to assess conflict management, so we need to interpret our findings with caution by also considering the risk of social desirability bias. Secondly, teachers enrolled in the MOOC on a voluntary basis, so we can hypothesize that the study sample could be less representative of those who were less open to diversity and, rather, consisted of people who were already disposed to achieve greater knowledge on the topic, and who were willing to enhance their management strategies for fostering positive relationships within their classes. The study is also limited considering that the final sample size is relatively poor. Nonetheless, the fact that 99 out of 206 teachers (about 48%) completed both the pre- and post-MOOC questionnaires should be considered a satisfactory result for many reasons. Indeed, whether these data represent the actual dropout of teachers, this percentage is still considerably low for a MOOC, which usually results in dropout rates over 90% (Goopio and Cheung 2021). Moreover, this number may not represent the people who left the MOOC but only those who did not complete the questionnaire. Also—and differently from e-learning courses—the full completion of the MOOCs, despite limiting the potential benefits deriving from the MOOC experience, is not the main goal/concern for several learners, who may indeed wish to learn new information/enhance their knowledge/face personal challenges (Breslow et al. 2013; Hew and Cheung 2014; Wang and Baker 2018). Finally, considering that the findings revealed no significant changes in the recourse to conflict management style by Gender, Age, and Working Seniority, we could sustain that the MOOC effectively produced changes in teachers beyond individual differences. However, given the small sample size, we do not mean to over-interpreting these data, which require further investigation on larger and more representative samples to be confirmed.

## 5. Conclusions

Nowadays, MOOCs represent the innovative node where the worlds of higher education, vocational education, training, and open online learning converge. By enhancing opportunities for the flexible delivery of education, tailored MOOCs—such as the one proposed in the present study—can effectively support new ways in which teachers can approach formal and informal education. Teachers can, indeed, achieve—in a flexible way—new knowledge and new skills, such as successfully dealing with conflicts and implementing/promoting inclusive educational practices.

From this perspective, the purpose of our tailored MOOC was to effectively help teachers to develop/enhance intercultural skills by offering innovative practices that promote

intercultural awareness and learning. Accordingly, we can confirm that this study provided evidence-based contributions which sustain the notion of the MOOC as a methodology that can significantly foster the awareness of teachers who wish to cultivate student–faculty relationships. The MOOC can promote a more effective form of intercultural and interethnic competency in classroom management.

Our findings can have several theoretical and practical implications. Specifically, considering theoretical implications, our study provides evidence which supports the usefulness of designing and developing effective learning strategies in MOOCs—that can be empirically measured and verified—in terms of perceptions of teachers regarding self-development in interethnic competencies. In other words, the present study endorses and tries to promote the development of research that empirically evaluates the effectiveness of online courses. To the best of our knowledge, the latter branch of this research is currently still lacking.

Considering practical implications, the MOOC that we have proposed is a multifaceted tool that teachers can use to train themselves—autonomously—but it can also be freely shared and used to train other colleagues and students. Indeed, the MOOC can be adopted as a culturally responsive teaching technique within educational practices to foster multicultural strength-based approaches. All key actors can be involved and expected to achieve results which are widely transferable within an interethnic context. Furthermore, considering the need for continuing professional development (CPD), teachers experiencing such MOOCs can take advantage of including a tailored multicultural training experience in their curriculum.

Finally, we consider that adopting tailored MOOCs covering culturally relevant issues—such as the one addressed in the present study—could indirectly impact society, fostering a better understanding of culturally and ethnically diverse populations, and creating welcoming communities. However, when designing multicultural training online, researchers/educators should directly involve, in the future, not only teachers but also educational stakeholders, students, and their families of different backgrounds. This approach could potentially have a greater impact on the current society, promoting inclusion, preventing conflict escalation, and spreading social justice. Promoting these key societal changes can start from the positive relational experiences within single classes. Yet, to address these challanges at a systemic level, it is crucial to spread these changes beyond the classroom walls to build a more inclusive and aware community.

**Author Contributions:** Conceptualization, E.D., F.V., M.C.Z. and D.M.; methodology, E.D. and F.V.; software, E.D. and D.M.; formal analysis, F.V.; investigation, E.D., F.V., M.C.Z. and D.M.; writing—original draft preparation, E.D. and F.V.; writing—review and editing, E.D. and F.V.; supervision, M.C.Z. and D.M.; project administration, D.M.; funding acquisition, E.D. and D.M. All authors have read and agreed to the published version of the manuscript.

**Funding:** This research received no external funding.

**Institutional Review Board Statement:** The study was conducted in accordance with the Declaration of Helsinki, and approved by the Institutional Review Board, Ethical Committee of Psychological Research (CERP) of University of Naples Federico II (Protocol Code: 36/2019; Approval Date: 28 November 2019).

**Informed Consent Statement:** Informed consent was obtained from all subjects involved in the study.

**Data Availability Statement:** The data presented in this study are available on request from the corresponding author. The data are not publicly available due to privacy.

**Conflicts of Interest:** The authors declare no conflict of interest. The funders had no role in the design of the study; in the collection, analyses, or interpretation of data; in the writing of the manuscript; or in the decision to publish the results.

## Appendix A

**Table A1.** The Five Lessons offered within the MOOC.

| Title | Description |
|---|---|
| Lesson 1.<br>Pedagogical framework | Competence areas: Intercultural literacy, Inclusive education; Conflict management |
| Lesson 2.<br>Multicultural Personality dimensions and teachers' role in multi-ethnic classes | Role of multicultural aspects of personality to cope with cultural diversity and to gain efficacy and success in multicultural contexts; relation to the school context, particularly focusing on the relevance of multicultural personality for teachers and educators. |
| Lesson 3.<br>Psychological models of conflict resolution in the school context | Negotiation as a dimension inextricably related to effective communication (assertive communication); Rahim's model of handling interpersonal conflict applied to the school context within teacher-pupil interethnic interactions. |
| Lesson 4.<br>The use of Serious Games in the school context for teacher-pupils' interethnic and intercultural interactions | Psychological modeling encompassing the game's development (integration and operationalization of the two main concepts of multicultural personality and Conflict Management Styles); main characteristics of the game and instructions for use. |
| Lesson 5.<br>Theoretical framing interethnic conflicts in the classroom: perspectives derived from sociology, anthropology, and social psychology | Concepts of ethnic inequality and ethnic discrimination in European educational systems; theoretical framing 'interethnic conflict' scenarios in the classroom. |

**Table A2.** The Serious Game scenarios within the MOOC.

| Title | Conflict Issue | Scenario Description |
|---|---|---|
| Scenario 1—<br>Too much noise | Majority vs. Minority cultures | A student of Ethiopian descent caught the teacher's attention during class because he was making too much noise. Despite several warnings from the teacher, the student and the teacher get caught up in a verbal conflict. At some point during the conflict, the boy screams to the teacher that 'history still lingers on' and 'white people still treat people of color as slaves'. |
| Scenario 2—<br>E-humor | Stereotyping and Ethnic humor | During an ICT-Class, a boy of Italian descent forwards an e-mail with a picture enclosed to his classmates. Suddenly, the whole class starts laughing. The picture shows a selfie of two monkeys wearing sunglasses and has a text: "this is a picture of us during our holidays in Brazil." He shouts: 'Sam's holiday picture [boy of African descent].' |
| Scenario 3—<br>Headscarf | Good intentions and Stereotyping | During class, the subject of ethnic discrimination comes up. The teacher invites everyone to share their experiences. One girl shares how people in the grocery store always look weird at her mother because she wears a headscarf. One of the class pupils states that people may not look odd at her because of what she wears but because of how she acts. Not everything that looks like a racist reaction at first sight is racist per se. The Muslim girl becomes furious and responds that this is not true. |
| Scenario 4—<br>Religions | Racism is exceptional | The subject of the course today is the exploration of religions. The teacher kindly asks a Muslim pupil to teach the class everything there is to tell about the religion of Islam. The teacher always enjoys it when her pupils can learn something new. The pupil refuses to accomplish the request of the teacher. |
| Scenario 5—<br>Movie Activity | Gender and Sexuality | The class watched a movie that tackled the theme of sexuality. One of the topics in the movie was homosexuality. After the movie, a discussion in class takes place. A Muslim boy of Turkish descent feels disgusted and shouts that two boys kissing should not be allowed and is completely unethical. The other classmates react and say that 'the Turkish is a backward culture and Islam is not a religion of modern times'. |

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
