# Peer review of "A Tailored MOOC Fostering Intercultural Conflict Management in the Educational Context: Evidence from Italy"

_socsci, doi:10.3390/socsci12060332_

Round 1

Reviewer 1 Report

In general, the text presents an interesting and novel idea, which may be attractive to educators who live similar realities.

However, there are several aspects that need to be worked on if the intention is to be published:

1. The theoretical section is very weak and contributes little to the study. First of all, it is necessary to include a section that presents the context of the study, that allows us to understand its relevance, why it occurs in this community and why it is an important topic at the educational level. This is not clear. In addition, it is necessary to include a complete section explaining the MOOC that was applied. Currently it includes information about MOOCs in general, but this does not mean that the MOOC used is so. We need to know what specifically was applied in the study.

2. It is necessary to make a table with the characteristics of the sample, especially to appreciate the variety of participants.

3. It is necessary to explain the reduction of participants. What exclusion criteria were used? Because there was a strong reduction and it is not understood why.

4. We need a table with the instrument to be able to see all the questions that were asked.

5. The methodology should include a section explaining the implementation. How was it done concretely with the teachers? What steps were followed?

6. The presentation of results is not attractive and very weak. Please strengthen it with some graphs that allow us to better appreciate the results, especially considering the characteristics of the population. It does not make sense that in the methodology they made the division by gender and age ranges if at the end they did not consider it in the analysis of results. I invite you to expand the analysis to allow us to appreciate all the possibilities of your study.

7. In the conclusions. What is your practical use of the results, how do they contribute theoretically and practically, and what lines of research do they open up? Conclusions should be improved

8. References need to be updated. It is necessary that at least half of them are from the last 5 years. 

9. The format of the letters should be checked since there is a mixture of formats in the references.

Author Response

Dear reviewer, thank you for your precious suggestions. Please see the attachment. 

Reviewer 2 Report

The article calls attention to a concern that has become more prevalent in classrooms across the globe, namely the issue as to how faculty members in university lecture halls have to shift away from traditional (i.e., inveterately existing) mindsets that favor optimal learning outcomes to more multifaceted perspectives that attempt to reconcile the desiderata of optimal learning outcomes and the cultivation of faculty-student relationships (lines 229~234). This core idea definitely deserves a scholarly article, and the present draft will fill a gap in the preexisting academic literature on educational practice and theory. The only suggestions that we wish to offer have to deal with the superabundance of quotations. We thank the authors for their due diligence in providing relevant literature on the topic, but we sometimes feel that these paraphrases and citations seem buried underneath the more argumentative things that the researchers should try to say. We also suggest that the authors look for overly laconic paragraphs (the one-sentence-long paragraphs of lines 30~32 and 235~241 immediately come to mind, but others surely exist) and lengthen these paragraphs with argumentative claims (and the elaborations of those claims alongside the conclusions of those paragraphs) that the authors wish to offer. In these elaborations, the authors should also aim for non-technical language without a reliance on theoretical terms infused with overly complicated ideas; an article overly burdened by theoretical language will become more difficult for readers to easily digest, since the argument would become buried under a superabundance of abstruse ideas that might seem unfamiliar to individuals outside the major field(s) of the co-authors.

The article should remove all instances of the eight passive verbs (am, are, be, been, being, is, was, were) in the article, unless the author(s) cite verbatim quotations that utilize those passive verbs. Scholars have usually had unflattering things to say about the usage of passive verbs in academic publications, since passive verbs unnecessarily lengthen the sentences of one's writing. T-Pronouns such as "this," "that," "those," and "these" should always precede actual nouns. Line 280, for instance, has a "this" reference without a noun following the word "this (In line with this, the dominating style may be inappropriately adopted by teachers 280 to aggressively impose his/her position along with depreciation, often without reasoning 281 and ignoring the needs and expectations of students )," so the reader has no idea about the reference for the word "this." In lines 317~320 and an infinity of other places, the authors have also used vague pronouns ("their," "its") that have zero clear antecedents. For instance, the sentences of lines 317~320 say, "In conclusion, this study provided evidence-based contributions sustaining that the MOOC we have proposed is able to significantly foster teachers' awareness and actual changes in their relational styles, thus supporting its effectiveness in enhancing intercultural and interethnic competencies in classroom management." However, the word "its" in "its effectiveness" can technically refer to the singular nouns "awareness," "MOOC," "study," or "conclusion," as all these singular nouns precede the word "its." Common sense would indicate the antecedent that the authors intend to use, but in the proper conventions of grammar, common sense serves as a very, very weak substitute for grammatical orthodoxy. The pronoun "their" also technically has two antecedents---namely the words "teachers" and "contributions," so the readers would have no clue about the antecedent. In other words, the authors should take care to absolutely remove all ambiguous pronouns for easier reading. Some sentences (like this one) also seem overly lengthy and could use some serious attempts at condensation. For the sentence of lines 317~320, the authors would immeasurably benefit by splitting the sentence into two parts and then deleting the ambiguous pronouns, resulting in something like the following: "In conclusion, this study provided evidence-based contributions sustaining the notion of the MOOC as a methodology that can significantly foster the awareness of teachers who wish to cultivate student-faculty relationships. The MOOC can therefore promote a more effective form of intercultural and interethnic competency in classroom management.

Author Response

(The authors gave the same response as above.)

Round 2

Reviewer 1 Report

Thank you very much, dear author, for your work. The truth is that the text is much more academically robust. You have improved the references, the theoretical framework is now much clearer and the methodology guides us through the implementation process. I consider that the conclusions are much more precise with what has been achieved and what is still pending.  I imagine it was laborious to adopt these suggested changes, but the current version shows a clear improvement. Thank you.